# Auto-inhibitory intramolecular S5/S6 interaction in the TRPV6 channel regulates breast cancer cell migration and invasion

Ruiqi Cai [1], Lingyun Wang[2], Xiong Liu [1], Marek Michalak [3], Jingfeng Tang[4✉], Ji-Bin Peng[2] & Xing-Zhen Chen [1✉]

TRPV6, a Ca-selective channel, is abundantly expressed in the placenta, intestine, kidney and bone marrow. TRPV6 is vital to Ca homeostasis and its defective expression or function is linked to transient neonatal hyperparathyroidism, Lowe syndrome/Dent disease, renal stone, osteoporosis and cancers. The fact that the molecular mechanism underlying the function and regulation of TRPV6 is still not well understood hampers, in particular, the understanding of how TRPV6 contributes to breast cancer development. By electrophysiology and Ca imaging in *Xenopus* oocytes and cancer cells, molecular biology and numerical simulation, here we reveal an intramolecular S5/S6 helix interaction in TRPV6 that is functionally autoinhibitory and is mediated by the R532:D620 bonding. Predicted pathogenic mutation R532Q within S5 disrupts the S5/S6 interaction leading to gain-of-function of the channel, which promotes breast cancer cell progression through strengthening of the TRPV6/PI3K interaction, activation of a PI3K/Akt/GSK-3β cascade, and up-regulation of epithelial-mesenchymal transition and anti-apoptosis.

[1] Department of Physiology, Faculty of Medicine and Dentistry, University of Alberta, Edmonton, AB, Canada. [2] Division of Nephrology, Department of Medicine, Nephrology Research and Training Center, University of Alabama at Birmingham, Birmingham, AL, USA. [3] Department of Biochemistry, Faculty of Medicine and Dentistry, University of Alberta, Edmonton, AB, Canada. [4] National "111" Center for Cellular Regulation and Molecular Pharmaceutics, Hubei University of Technology, Wuhan, Hubei, China. ✉email: tangjingfeng@hbut.edu.cn; xzchen@ualberta.ca

The mammalian transient receptor potential (TRP) super-family of ion channels comprises 28 members that are involved in sensory processes through regulation by diverse chemical, mechanical, or optical stimuli[1]. TRP vanilloid 6 (TRPV6), with its closest homolog TRPV5, is the most Ca-selective channel within the TRP superfamily[2]. When initially cloned from the rat duodenum, TRPV6 was proposed to mediate transcellular Ca absorption in epithelial cells of the small intestine[3]. TRPV6 is detected in the kidney, pancreas, mammary gland, salivary gland, sweat gland, and reproductive organs[2]. Abnormal expression or function of TRPV6 is linked to male fertility, transient neonatal hyperparathyroidism, kidney stone formation, and carcinoma[2] but the underlying mechanisms have remained largely unclear.

Recent technical advances in cryogenic electron microscopy (cryo-EM) allowed revealing high-resolution structures of TRPV6 and other TRP channels in open and closed states[4]. Similar to other TRP channels, TRPV6 has cytosolic N- and C-termini and six membrane-spanning domains, with aspartic acid residue 582 (D582) in the loop between helices S5 and S6 as part of its selectivity filter. The four S6 helices in the tetrameric TRPV6 channel form the pore-lining region, in which methionine 618 (M618) was suggested by a closed TRPV6 cryo-EM structure to form a hydrophobic energy barrier to ion flow[4]. Interestingly, a functional study instead found that the pore gate is formed by consecutive residues A616 and M617[5]. The S6 helix was reported to undergo transition from an α helical configuration in the closed state to a π helical configuration in an open state, which enlarges the pore gate. We recently identified functionally auto-inhibitory intramolecular interactions of the S4–S5 linker and pre-S1 domain with the TRP domain in TRPV6 and found that each interaction is mediated by a conserved residue pair and negatively regulated by phosphatidylinositol 4,5-bisphosphate (PIP2)[6]. TRPV6 channel function substantially increased when either interaction was disrupted[6]. Structures of TRPV6 suggest the presence of more intramolecular interactions, which have yet to be investigated and may be relevant to TRPV6-dependent physiology and pathology.

Breast cancer is globally the most common and fatal form of cancer in women[7]. Depending on the presence or absence of estrogen receptor (ER), human epidermal growth factor receptor 2 (HER2) and progesterone receptor (PR), breast cancer is clinically classified into luminal A (ER+/HER−/PR+), luminal B (ER+/HER+/PR−), HER2 positive (ER−/HER+/PR−), and basal-like (ER−/HER−/PR−) to guide the hormone therapy targeting the present receptor. The basal-like breast cancer, also known as triple-negative breast cancer, is the deadliest form and currently has the poorest treatment outcome. Previous studies found elevated mRNA and protein expression of TRPV6 in tumors, compared with the normal breast tissue[8]. Subsequent studies found enrichment of the TRPV6 protein in invasive regions of breast tumor tissues[9]. Further, TRPV6 knockdown or use of inhibitor to TRPV6-expressing MCF-7, MDA-MB-231, and T-47D cell lines impeded cell viability, migration, and/or invasion[9,10]. These studies indicated the relevance of TRPV6 GOF or, more specifically, TRPV6-mediated increase in the Ca flow, to breast cancer development. The mechanism of how TRPV6 promotes breast cancer progression has yet to be determined.

Cancer cells are characterized by elevated proliferation, migration, invasion, and apoptotic resistance. One of the known pathways controlling these processes depends on Akt/GSK-3β, which also regulates the stability and expression of the epithelia-mesenchymal transition (EMT) and prosurvival markers. The class IA phosphoinositide 3-kinase (PI3K) as a heterodimer is composed of the regulatory subunit p85 and catalytic subunit p110[11] and acts as an upstream activator of Akt/GSK-3β. p85 stabilizes and inhibits p110 via direct binding. PI3K can be recruited to near the surface membrane and activated by receptors or adaptor molecules[12]. Upon extracellular stimuli or in the presence of mutations in oncogenes, epithelial cells may undergo EMT to attenuate cell–cell adhesion thereby becoming highly mobile and invasive. Various ion channels, including Cl, K, Na, and Ca channels, were found to implicate in the EMT in breast, liver, lung, and head/neck cancers[13]. In particular, Ca channels Orai1, TRPC6, -M7, and M8 have been proposed to induce the EMT in breast or liver cancer cells[13]. Whether and how TRPV6 is involved in EMT remains unclear.

Our present study utilized the two-electrode voltage clamp in *Xenopus* oocyte, Ca imaging in cultured breast cancer cells MCF-7 and MDA-MB-231, molecular biology, and molecular dynamics (MD) simulation to investigate a functionally autoinhibitory intramolecular interaction between TRPV6 helices S5 and S6 and how GOF of TRPV6 and pathogenic mutation R532Q contribute to breast cancer cell progression through a PI3K/Akt/GSK-3β pathway. It discussed significance of these gained mechanistic insights into TRPV6 gating and TRPV6-dependent human breast cancer progression to therapeutic interventions.

## Results

### Functional importance of the conserved residue R532 in helix S5.
While the S4–S5 linker in TRPV6 is a critical regulatory element for its channel function[14] the functional of helix S5 which is in close proximity to the pore-forming S6 as revealed by structures[15] is not well understood. Among the three conserved positively charged residues (K524, R510, and R532) in the S4–S5 linker or S5 helix, R510 mediates the S4–S5 linker/TRP domain binding, K524 is critical for TRPV6 activation by PIP2[6,16], but the role of R532 in S5 is unclear. Interestingly, the Catalogue of Somatic Mutations in Cancer (COSMIC) database predicted R510Q and R532Q as pathogenic mutations[17]. Here by use of the two-electrode voltage clamp electrophysiology in *Xenopus* oocytes as an expression model we found that, compared with wildtype (WT) TRPV6, the TRPV6 R532Q mutant is of substantial GOF, with a 50-fold increase in the currents induced by 5 mM extracellular Ca (Fig. 1a) but unaffected surface membrane expression (Figs. 1b and S1), indicating the functional importance of R532. Of note, these and other similar experiments in oocytes were carried out in the presence of 10 mM Cl channel blocker 2-[(4-methoxynaphthalen-2-yl)amino]-5-nitrobenzoic acid (MONNA) to minimize the Cl currents mediated by endogenous Cl channels in oocytes. To further dissect the role of R532 we mutated R532 to aromatic or charged residues and found that, similar to R532Q, substitution of R532 with aromatic or negatively charged residues results in substantially increased Ca-induced currents (Fig. 1a, b). In contrast, the channel activity of mutant R532K resembled that of WT TRPV6, indicating the importance of a positive charge at site 532 to maintain the TRPV6 basal function.

### Role of the R532:D620 pair in mediating the S5/S6 helix interaction.
We next wanted to determine whether R532 inter-acts with another residue to confer its functional role. Based on physical proximity of R532 with residues in helix S6, as revealed by TRPV6 structures, we wondered whether the highly conserved aspartic acid 620 (D620) in the distal part of S6 can interact with R532 (Fig. S2). Indeed, removal (D620A) or reverse (D620R) of the negative charge in D620 resulted in similar GOF as R532Q (Fig. 1c, d). Importantly, substitution of D620 with aromatic tryptophan (W) or phenylalanine (F) or with glutamic acid (E) all resulted in channel activities similar to that of the WT channel (Fig. 1c, d), indicating the possibility that these substitutions

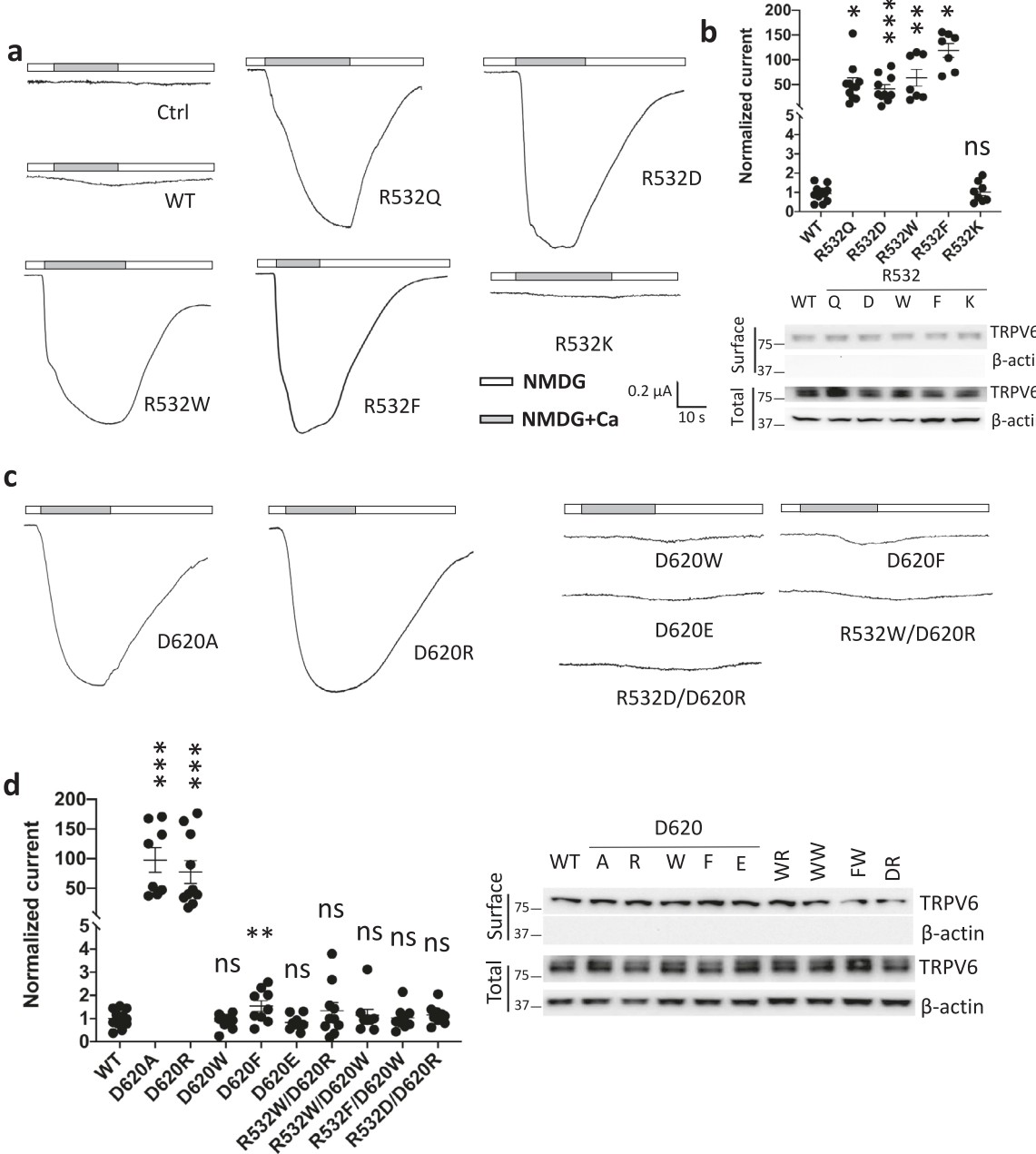

**Fig. 1 Functional roles of the TRPV6 residues R532 and D620. a** Representative current traces from water-injected (Ctrl) oocytes and those expressing WT TRPV6 or a R532 mutant. **b** Upper panel: averaged and normalized current values for WT TRPV6 WT or a R532 mutant expressed in oocytes. ***$p < 0.001$; ns not significant. Lower panel: total and surface expression of WT TRPV6 and R532 mutants revealed by biotinylation. β-actin acts as loading control. **c** Representative current traces from oocytes expressing different TRPV6 mutants. **d** Left panel: averaged and normalized current values corresponding to WT or mutant TRPV6. Right panel: total and surface expression of WT or mutant TRPV6. **$p < 0.01$; ***$p < 0.001$; ns not significant, from three independent experiments.

result in the formation of a cation–π bond (R:W or R:F) or salt bridge (R:E) between sites 532 and 620 and therefore maintained the basal channel activities. To provide further evidence that site 532 in S5 may interact with site 620 in S6, and to examine whether GOF mutations are due to protein misfolding, we carried out experiments to determine whether we could 'rescue' GOF mutants. For this purpose we introduced a second mutation (D620R or D620W) to GOF single mutants R532W, R532D, and R532F and found that the resulting double mutants R532W/D620R, R532W/D620W, R532D/D620R, and R532F/D620W all exhibited similar activities as WT TRPV6, i.e., either of the

second mutations rescued the GOF of the single mutants (Fig. 1c, d), presumably because new bonds (W:R, W:W, D:R, and F:W) were re-established at 532:620 in double mutants R532W/D620R, R532W/D620W, R532D/D620R, and R532F/D620W, respectively. Note that the conclusion drawn above made use of the fact that the surface and total protein expressions were not affected by any of these mutations (Figs. 1d and S3). Consistently, Na currents mediated by R532Q and D620A both showed a significant increase compared to that by WT channel (Fig. S4). 2-APB as a known TRPV6 inhibitor was examined for its inhibition of mutant channels R532Q and D620A. We found that 2-APB

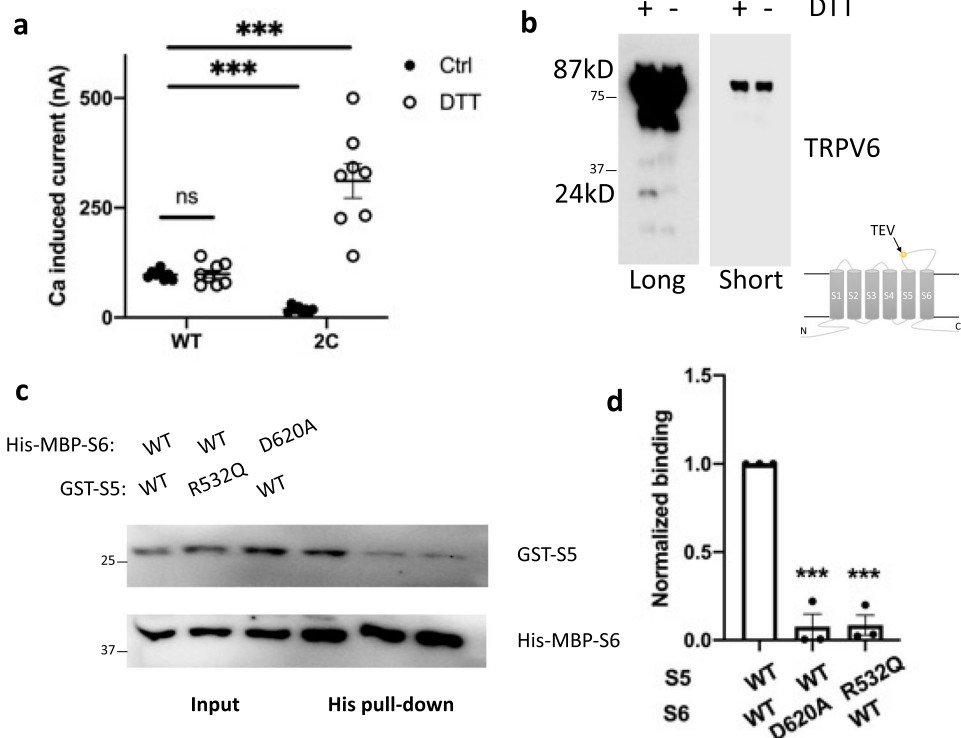

**Fig. 2 Associations between R532 and D620 in TRPV6.** **a** Statistical analysis of currents from oocytes expressing TRPV6 or mutant R532C/D620C, with or without 5 µM DTT. ***$p < 0.001$; ns not significant. **b** Representative WB images with long (long) or short (short) exposure time showing the DTT effect on mutant R532C/D620C$_{TEV}$. illustration indicating the insertion site of the TEV cleavage. **c** Representative images on interaction between purified S5 and S6 fragments, without or with an indicated mutation. **d** Averaged and normalized S5/S6 helix binding, quantified using ImageJ. ***$p < 0.001$, from three independent experiments.

inhibits the two mutants and WT TRPV6 with similar potencies (Fig. S5).

We next wondered whether a covalent disulfide bond can be introduced at 532:620, which would be stronger than the bonds described above and can be regulated by the reducing reagent dithiothreitol (DTT). Double mutant R532C/D620C exhibited significantly smaller channel activity than the WT channel (Fig. 2a), which is consistent with the formation of a disulfide bond at 532:620 that is stronger than an electrostatic salt bridge in WT TRPV6. Incubation with DTT significantly increased the R532C/D620C channel activity but had no effect on WT TRPV6 (Fig. 2a), indicating the effectiveness of the DTT treatment. We next wanted to document the importance of the 532:620 bonding for the S5/S6 helix association that could also be mediated by other residue pairs. For this we inserted a tobacco etch virus (TEV) protease cleavage sequence after E559 in the extracellular S5–S6 loop of mutant R532C/D620C and name the resulting mutant as R532C/D620C$_{TEV}$. Application of TEV is expected to generate a 24 kD fragment containing the C-terminus, which should be detectable by our TRPV6 antibody. Indeed, a fragment of around 24 kD was detected in the presence of DTT but not in its absence using oocytes expressing mutant R532C/D620C$_{TEV}$ treated with the TEV protease (Figs. 2b and S6), which indicates that the disulfide bond formed at 532:620 is key to the S5/S6 association. We also carried out pull-down assays to determine whether the S5/S6 helix interaction is direct. For this we purified the fragments of S5 (G511-V540) and S6 (A603-L632) and indeed detected binding of GST-G511-V540 with His-MBP-A603-L632, which was disrupted in the presence of mutation R532Q or D620A in one or the other fragment (Fig. 2c, d). Of note, as GST or His-MBP do not bind with the TRPV6 S5 or S6 fragment (Fig.

S7), these data demonstrate that direct S5/S6 helix binding is mediated by R532:D620.

## The R532:D620 interaction investigated by MD simulation.
In the initial MD simulation model there was no hydrogen bond formed between R532 and D620 (Fig. S8a). To test whether it is the case during the simulation, the formation of a hydrogen bond between residues R/Q532 and D620 was calculated. We found that a hydrogen bond is indeed formed between R532 and D620 within each monomer of WT TRPV6 during the simulation while no hydrogen bond was formed between Q532 and D620 in mutant R532Q (Fig. 3a). Compared with WT TRPV6, the final structures for mutant R532Q showed that residue D620 undergoes much larger position changes (Fig. 3b), suggestive of absent Q532:D620 binding. These results are thus consistent with the 532:620 interaction revealed by means of molecular biology and electrophysiology (Figs. 1 and 2).

Since D620 is close to the gate residue M618 (McGoldrick et al., 2018) we reasoned that structural and dynamic changes in D620 may induce corresponding changes in M618. To test this proposition, we calculated the helix structure occupancy and root mean square fluctuation (RMSF) for each of the two systems (with and without the R532Q mutation). Compared with the WT TRPV6 system, the helix occupancy and RMSF for D620 in the R532Q system was significantly decreased and increased, respectively (Fig. 3c, d), indicating that the R532Q mutation reduced the structural and dynamic stability of D620. Detailed analyses also showed that the R532Q mutation similarly alters the secondary structure and dynamic stability of a region around D620 (M618–W623).

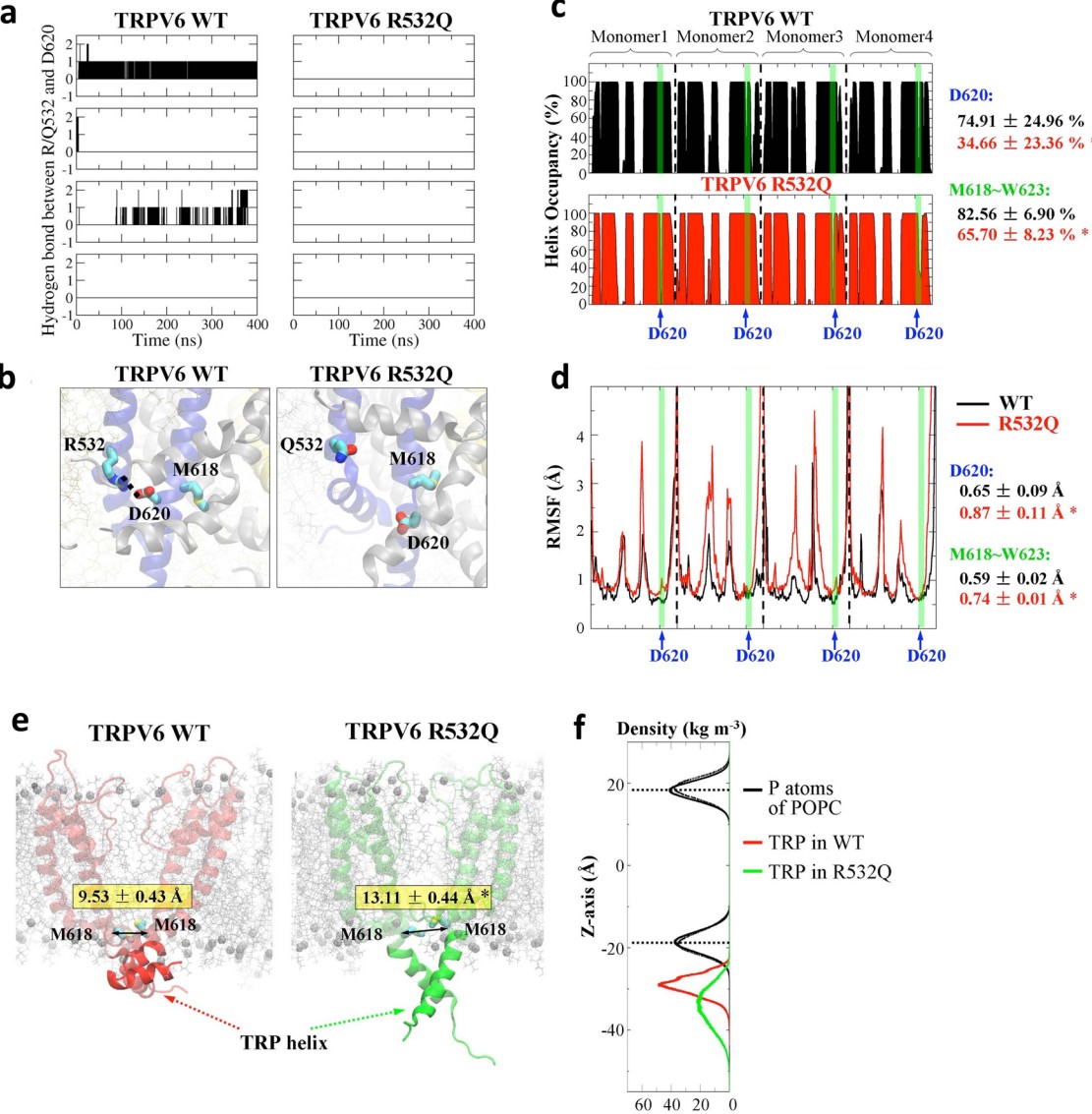

**Fig. 3 R532:D620 interaction examined by MD simulation. a** Hydrogen bonding between a residue at 532 (R532 or Q532) and D620 as a function of time. **b** Effect of the R532Q mutation on the D620 conformation. Structural (**c**) and dynamic (**d**) changes in D620 and its proximal region (M618–W623). The red and black lines show the RMSD values of TRPV6 with and without the R532Q mutation, respectively, during the 400 ns simulations. A higher occupancy value means a more stable secondary structure, while a higher RMSF value indicates higher dynamic flexibility of the residue. **e** Effect of the R532Q mutation on the distance between the two diagonal M618 in a TRPV6 tetramer. **f** Effect of the R532Q mutation on the mass density of phosphate (P) atoms in POPC lipid (black curves) and the TRP helix of TRPV6 along the Z-axis. The membrane surface positions are indicated by the density peaks of lipid P atoms (dotted lines). Note: for a display purpose, shown density values of POPC P atoms were reduced by ×10 from the real values.

To further test whether the R532Q mutation causes dynamic changes in gate residue M618 we calculated the distance between the two diagonal M618 in a TRPV6 tetramer (Fig. 3e). We found that the mutation causes a significant increase in the distance ($9.53 ± 0.43$ Å in WT TRPV6 vs. $13.11 ± 0.44$ Å in mutant R532Q). Compared with the initial model (12.46 Å, Fig. 3b), the distance was significantly decreased in WT TRPV6 but not in mutant R532Q. Thus, the channel pore (defined by the four gate residues M618) was stabilized from 12.46 Å to a smaller size (9.53 Å) in WT TRPV6, which nice accounts for the relatively small channel activity of WT TRPV6 at basal states, while the pore size of the mutant channel remained large (13.11 Å), consistent with the GOF. In addition, we found that the TRP helix in TRPV6 R532Q undergoes dramatical conformational changes, which were not seen in the WT channel (Fig. 3e). To

better illustrate these changes we calculated the density for the TRP helix to indicate the TRP helix position along the Z-axis. We found that the TRP helix in WT TRPV6 is located near the surface membrane (red line, Fig. 3f), while that in the mutant is located far away from the surface (green line, Fig. 3f), indicating its much higher mobility in the mutant.

In summary, our MD simulation revealed the formation of a hydrogen bond between R532 and D620 in WT TRPV6, which allowed stabilizing the structure and dynamics of D620 its proximal region comprising gate residue M618, thereby probably stabilizing the channel pore to a small size. The R532Q mutation may have disrupted the structural and dynamic stability of the pore gate and presumably induced large conformational changes in the TRP helix, which together would account for the GOF of the mutant channel.

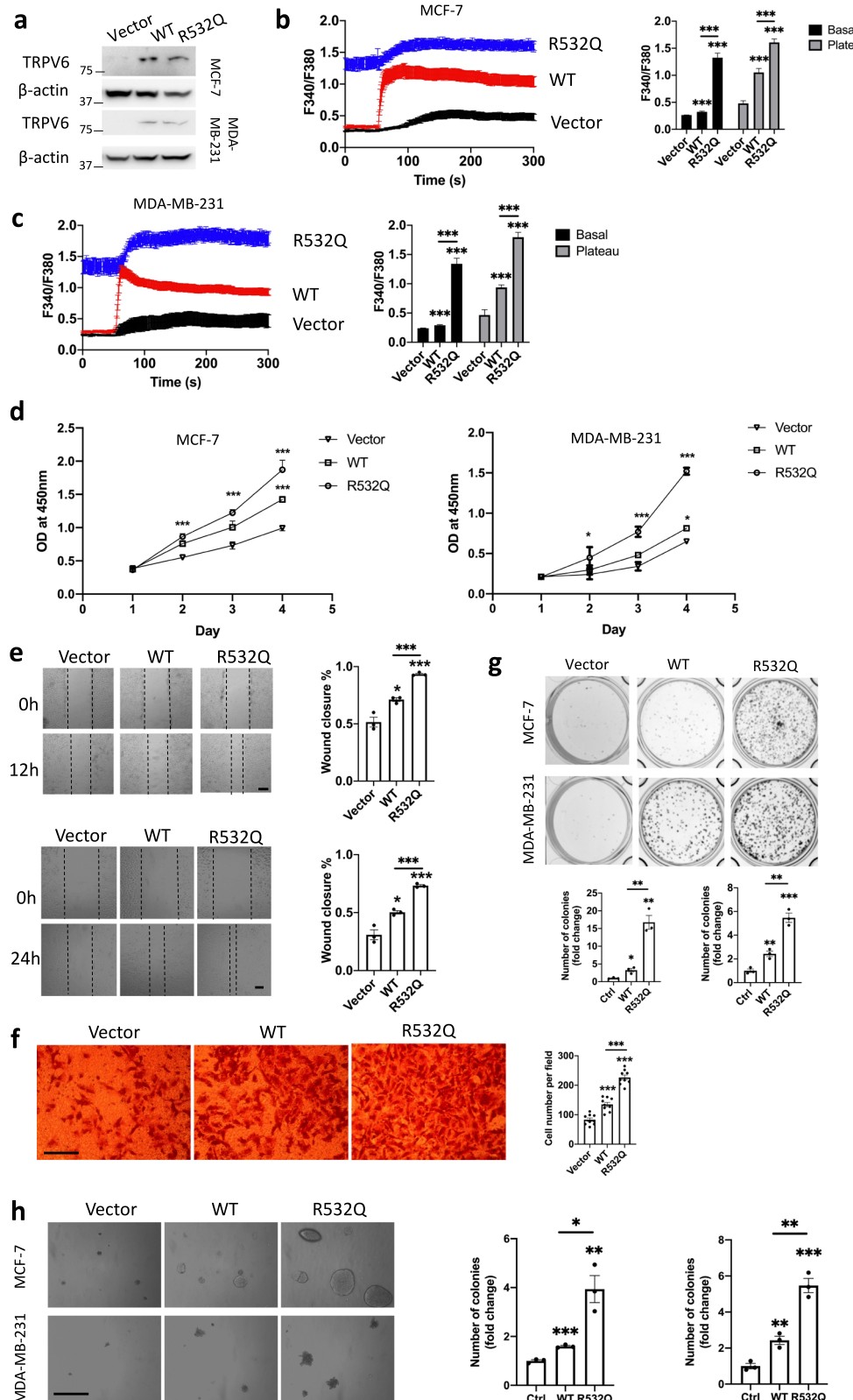

**Effect of mutation R532Q on breast cancer cell progression**. The expression of transfected WT TRPV6 and mutant R532Q in both MCF-7 and MDA-MB-231 cell lines was confirmed by western blotting (WB) (Fig. 4a), with endogenous TRPV6 detectable through immunoprecipitation followed by immunoblotting[18]. We then utilized Ca imaging to the cell lines to determine the effect of the TRPV6 channel function on breast cancer cell growth. We found that, compared with vector transfected (control) cells, TRPV6-expressing cells have significantly higher basal and plateau cytosolic Ca levels before and after the application of 2 mM extracellular Ca, respectively (Fig. 4b, c). Further, cells expressing mutant R532Q exhibited even higher cytosolic Ca levels than those expressing WT TRPV6 (Fig. 4b, c). Note that the higher basal Ca level may be due, at least in part, to the Ca contained in the culture

**Fig. 4 Effects of the R532Q mutation and TRPV6 channel function on breast cancer cell progression. a** Representative expression of TRPV6 and mutant R532Q stably expressed the two cell lines revealed by WB. **b** Ca imaging experiments in MCF-7 stable cell line. Left panel: cytosolic Ca measured by Fura-2 AM based Ca imaging, before and after application of 2 mM extracellular Ca. Traces are averaged from 15–19 measurements of three independent experiments. Right panel: statistical analysis of averaged basal Ca (0–50 s) and plateau Ca (250–300 s) concentrations. ***$p < 0.001$. **c** Ca imaging experiments in MDA-MB-231 stable cell line performed under similar conditions as in **b**. **d** Cell proliferation assays in MCF-7 and MDA-MB-231 stable cell lines. *$p < 0.05$, ***$p < 0.001$. **e** In vitro scratch experiments to evaluate cell migration. Representative images and statistical data showing migration of MCF-7 (upper) and MDA-MB-231 (lower) stable cell lines. Scale bar 200 μm. *$p < 0.05$, ***$p < 0.001$. **f** Representative images and statistical data showing the effect of the R532Q mutation and TRPV6 expression on the colony formation of MCF-7 and MDA-MB-231 stable cell lines. **g** Representative images and statistical data showing the effect of the R532Q mutation and TRPV6 expression on the invasion of MDA-MB-231 stable cells. Scale bar 100 μm. **h** Representative images and statistical data on the anchorage-independent colony formation of MCF-7 and MDA-MB-231 stable cell lines in soft agar. Scale bar 200 μm. *$p < 0.05$; **$p < 0.01$; ***$p < 0.001$, from three independent experiments.

medium. These data are in support of the GOF nature of the R532Q mutant revealed by means of electrophysiology (Fig. 1) and displayed positive correlation between the TRPV6-mediated Ca entry and cytosolic Ca level.

Malignant cancer cells are characterized by enhanced proliferation, migration, invasion as well as ability to form colonies from individual cells[19]. We found that the proliferation of cells expressing mutant R532Q and WT TRPV6 was dramatically and moderately increased, respectively, compared with those transfected with empty vector (Fig. 4d, e). By in vitro scratch assays we found that the effects of mutant and WT TRPV6 on cell migration are substantial and moderate (but significant), respectively (Fig. 4e). Given that MDA-MB-231, but not MCF-7, cells are invasive we examined the invasion of these cells and found that cells expressing R532Q are more invasive than those expressing WT TRPV6, which are more invasive than control cells (Fig. 4f). We next examined the oncogenicity of MCF-7 and MDA-MB-231 cells by colony formation and soft agar assays. Expression of R532Q and WT TRPV6 had pronounced and mild (but significant) stimulating effects, respectively, on the number of colonies (Fig. 4g, h), which was similar to its effects on proliferation, migration, and invasion. These data together demonstrated that mutant R532Q has stronger stimulating effects on breast cancer cell progression than WT TRPV6, which has a moderate stimulating effect.

**Involvement of a PI3K/Akt/GSK-3β cascade and the cytosolic Ca in the regulation of EMT and antiapoptosis of breast cancer cells by mutation R532Q.** We next investigated how TRPV6 promotes cancer cell progression. We found that the p-Akt level is highest in R532Q-expressing cells and is significantly higher in TRPV6-expressing cells than control cells, which were supported by our consistent data on the effects of R532Q and WT TRPV6 on GSK-3β (Fig. 5a–c), knowing that Akt phosphorylation/activation leads to GSK-3β inhibition/phosphorylation (at Ser9)[20]. We further revealed changes in the expression of Snail, E-cadherin and vimentin due to expression of R532Q or WT TRPV6 (Fig. 5a–c). These data strongly indicated the involvement of a PI3K/Akt/GSK-3β pathway in the regulation of cancer progression by TRPV6 because Snail as an EMT marker inhibitable by the GSK-3β activity is known to suppress E-cadherin and promote the vimentin expression[21,22]. Also consistently, p-GSK-3β increases the expression of Mcl-1, an antiapoptotic marker (Fig. 5a–c) and an important factor for cell viability.

We also examined how Ca chelator BAPTA-AM or EGTA alters the effects of TRPV6 on the signaling cascade. We found that treatment with BAPTA-AM or EGTA abolishes the effects of WT TRPV6 but not those of mutant R532Q (Figs. 5d, e, S9 and S10). In contrast, treatment with PI3K inhibitor LY294003 completely abolished all the effects of R532Q (Figs. 5d, e and S9). In summary, our data demonstrated that GOF of TRPV6 as a Ca

channel strongly promotes EMT and antiapoptosis through a PI3K/Akt/GSK-3β pathway.

**Interaction of TRPV6 with p85.** The inhibitory subunit p85 of PI3K can be recruited to the surface membrane through interaction with various membrane proteins and then activates the catalytic subunit p110 of PI3K[23]. TRPV6 stably expressed in MDA-MB-231 cells interacted with endogenous p85 in a Ca-dependent manner and the interaction was enhanced by the R532Q mutation (Figs. 6a, b and S11). Consistently, the mutation significantly enhanced recruitment of p85 to the surface membrane by TRPV6 (Figs. 6c and S12). To dissect the p85 domain responsible for the TRPV6/p85 interaction we generated four truncation mutants for co-immunoprecipitation (co-IP) assays and found that both the nSH2 and cSH2 domains in p85 and the TRPV6 N-terminus (TRPV6N) mediate the interaction (Fig. 6d, e). Using purified fragments of p85 and TRPV6N for in vitro binding we found direct TRPV6N/nSH2 and TRPV6N/cSH2 association (Fig. 6f). To examine the role of the TRPV6/p85 interaction in the PI3K/Akt/GSK-3β-dependent signaling we expressed the p85 fragment nSH2 as a potential blocking peptide to competitively disrupt the TRPV6/p85 interaction. Indeed, expression of nSH2 abolished the stimulation of the EMT markers and antiapoptotic Mcl-1 by both WT and mutant TRPV6 (Figs. 6g and S13). Thus, TRPV6, presumably its channel function, upregulates EMT and apoptosis resistance through direct binding with both the nSH2 and cSH2 domains of p85, thereby activating the PI3K/Akt/GSK-3β cascade.

**Correlation between the TRPV6 expression and poor survival of breast cancer patients.** It was previously reported that patients with ER− breast cancer have higher TRPV6 mRNA expression than ER+ patients[10]. We here analyzed possible relationship between the TRPV6 mRNA expression and survival rate of breast cancer patients using published breast cancer patient databases[24] to explore the clinical relevance of our in vitro study. For all breast cancer types, our analysis showed significant inverse correlation of the TRPV6 mRNA level with the overall survival rate (HR = 1.58, 1.25–1.99; $p = 8.8e-05$) and the relapse-free survival rate (HR = 1.45, 1.29–1.63; $p = 6.9e-10$), with a 95% confidence interval (Fig. 7a, b). We also analyzed the expression of *TRPV6* in ER− breast cancer patients and found that the high *TRPV6* level is associated with poor overall survival (HR = 1.92, 1.25–3.04; $p = 0.0042$) and relapse-free survival (HR = 1.31, 1.04–1.64; $p = 0.019$) (Fig. 7a, b). Thus, these significant inverse correlations between the *TRPV6* expression and survival rates found in all and ER− breast cancer patients nicely supported our data obtained using in vitro systems.

**Discussion**

As one of the most highly Ca-selective channels in the TRP superfamily TRPV6 has been known for its importance for

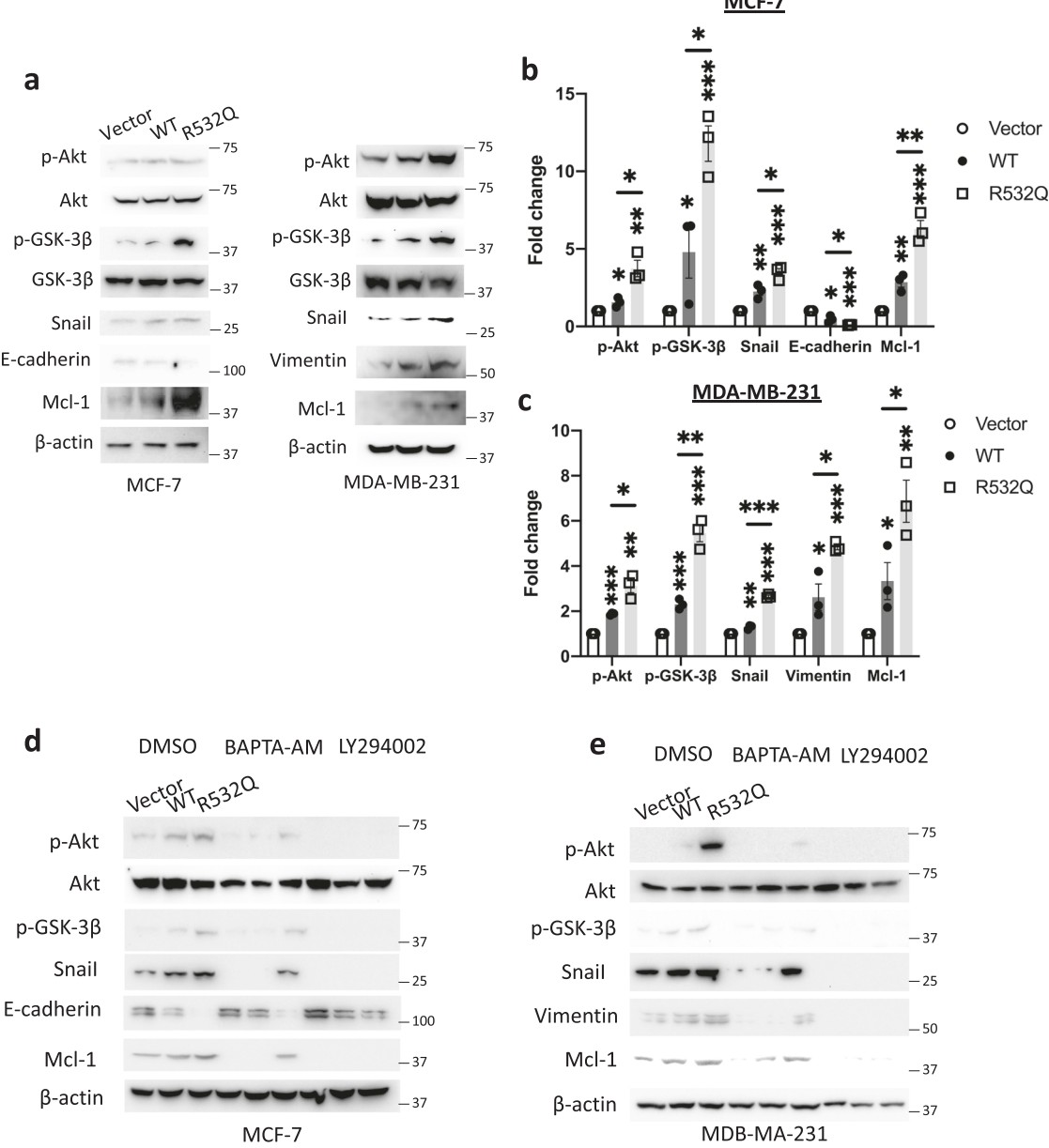

**Fig. 5 Involvement of a PI3K/Akt/GSK-3β pathway in and effect of the TRPV6 function on breast cancer development. a** Representative WB data showing changes in EMT and antiapoptotic markers in MCF-7 (left) and MDA-MB-231 stable cell lines. Statistical analysis of WB data of the expression of indicated markers in MCF-7 (**b**) and MDA-MB-231 (**c**) stable cells. **d**, **e** Effects of BAPTA-AM (10 μM, 1 h) and PI3K inhibitor LY294002 (50 μM, 1 h) on the EMT and antiapoptosis markers, with DMSO treatment as control. Shown are representative WB data obtained using MCF-7 (**d**) and MDA-MB-231 (**e**) stable cells. *$p < 0.05$; **$p < 0.01$; ***$p < 0.001$, from three independent experiments.

mediating transcellular Ca absorption in multiple tissues. Malfunction due to genetic mutations in or altered expression of TRPV6 has been clinically linked to human breast cancer and several other diseases. However, how TRPV6 channel is gated and exercises its oncogenic effects are not well understood. By electrophysiology, Ca imaging, molecular biology and MD simulation, our present study identified an autoinhibitory intramolecular S5/S6 helix interaction and showed that this interaction is mediated by the R532:D620 pair of residues located in proximal S5 and distal S6, respectively. Using MDA-MB-231 and MCF-7 human breast cancer cell lines we demonstrated the oncogenicity of WT TRPV6 and GOF mutant R532Q predicted to be pathogenic. Specifically, we found that the TRPV6 channel function promotes the viability, migration and invasion via

increased cytosolic Ca and interaction with p85, which activates a PI3K/Akt/GSK-3β pathway, EMT, and antiapoptosis.

TRP channels are tetramers featuring domain swapping also seen in voltage-gated ion channels[25], i.e., the S5 and S6 helices of a monomer are in contact with the S1–S4 helices of an adjacent monomer. TRPV6 structure suggested possible presence of a hydrogen bond formed between D529 in S5 and T621 in S6[15] and that mutation T621A be associated with loss-of-function[26]. Thus, the D592:T621 interaction, which was not observed in the closed state structure of TRPV6 channel[15], was thought to maintain an open state of TRPV6, but with little functional evidence. Distinct from this concept about the D529:T621 interaction, we found that the S5/S6 helix interaction mediated by R532:D620 is autoinhibitory for TRPV6 channel function, i.e.,

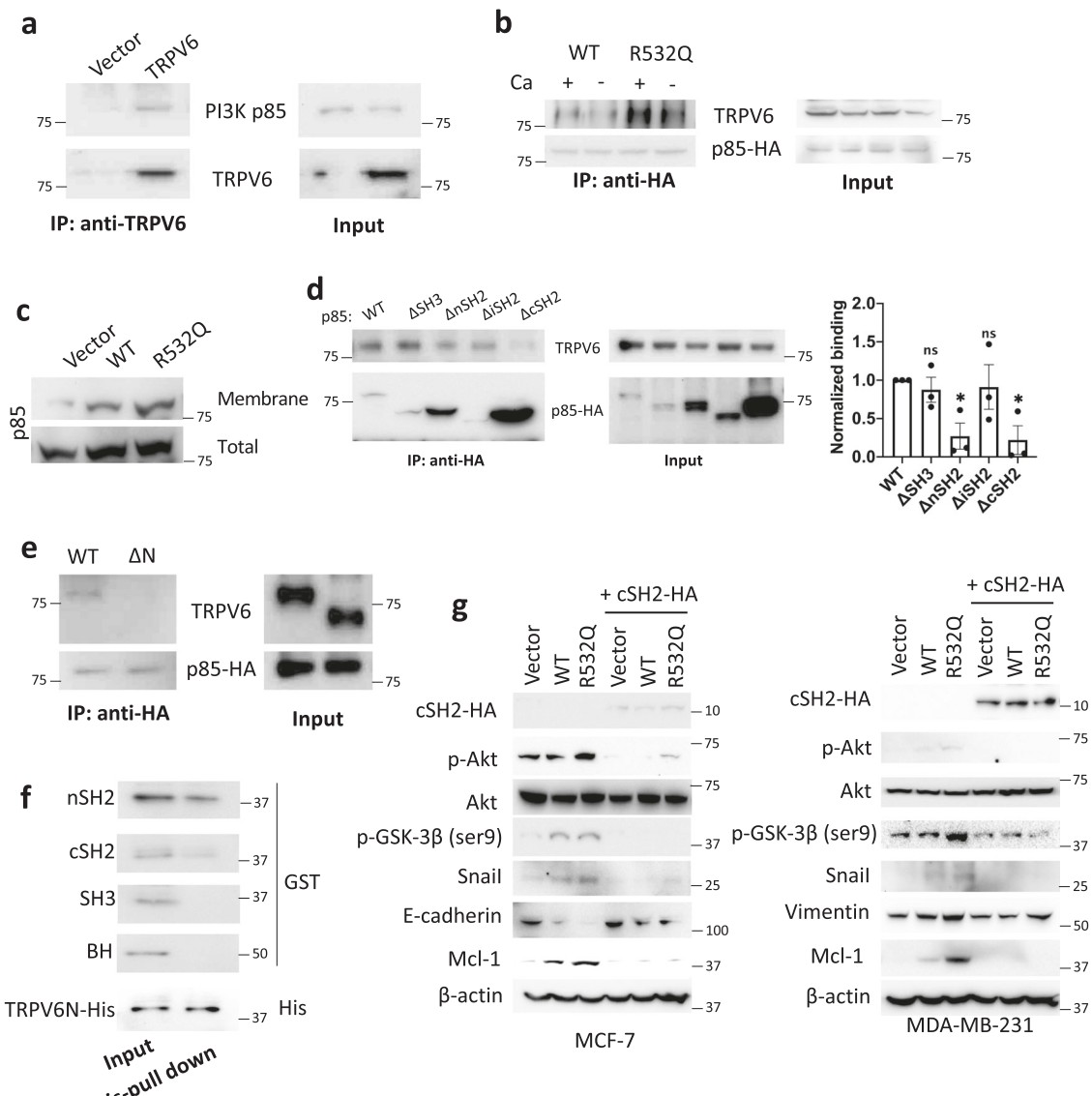

**Fig. 6 The TRPV6/p85 interaction and its role in the activation of a PI3K/Akt/GSK-3β pathway by TRPV6 and mutant R532Q in breast cancer cells.**
**a** Representative co-IP data showing interaction of TRPV6 with endogenous p85 in MDA-MB-231 cells. **b** Representative co-IP data showing the effect of mutation R532Q on the interaction of TRPV6 with p85-HA without (1 mM EGTA) or with (0.1 mM) Ca using MDA-MB-231 stable cells. **c** Effect of mutation R532Q on the total and membrane bound p85 in MDA-MB-231 stable cells. **d** Interaction of transiently expressed HA-tagged p85 truncation mutants with stably expressed TRPV6 by co-IP in MDA-MB-231 cells. ΔSH3, ΔnSH2, ΔiSH2, and ΔcSH2 indicate p85 with deletion of SH3 (S2-R93), nSH2 (M322-K430), iSH2 (D434-A614), and cSH2 (L617-Y724), respectively. **e** Representative co-IP data showing interaction of WT TRPV6 WT and its truncation mutant ΔN (TRPV6 with deletion of the N-terminus, Y328-I725) with p85 in transiently transfected MDA-MB-231 cells. Right panel: Statistical data from three independent experiments showing normalized binding strengths. **f** Representative in vitro binding data showing binding of purified TRPV6N-His with p85 fragments, as indicated. **g** Representative WB data showing the blocking peptide effect of transiently expressed cSH2-HAon activation of EMT and antiapoptosis markers by stably expressed TRPV6 and its mutant R532Q in MCF-7 and MDA-MB-231 cells. *$p < 0.05$; ns not significant. All data were from three independent experiments.

the R532:D620 bonding is to maintain the basal state and its disruption leads to GOF of the channel. In fact, our finding is consistent with how the R532:D620 pair distance (D) changes between states based on TRPV6 structures, which revealed that $D_{open} > D_{closed} > D_{2\text{-}ABP\ inhibited}$ (Fig. S14). Examining the corresponding potential S5/S6 helix interactions in TRPV1, -V2, and -V3 structures at apo or ligand-bound (open/activated) states revealed a similar relationship between the distance and channel function, i.e., $D_{open/ligand\text{-}bound} > D_{closed/apo}$ (Fig. S14). Of note, the S5/S6 helix interaction was not observed in the 2-APB-bound TRPV6 structure (Fig. S14) presumably because the distances

between R532 and D620 is 4.6 Å, which is larger than the maximum distance (4 Å) for forming a salt bridge[27]. However, TRPV6 proteins expressed on live-cell membranes should have different structural configurations than those obtained under cryo-EM conditions and thus may have the R532:D620 distance less than 4 Å at inhibitor-bound or closed states, i.e., may indeed form a salt bridge at R532:D620. Our concept of a 532:620 bonding is strongly supported by our functional analyses using electrophysiology, physical interaction, and MD simulation in the presence of various residue substitutions at R532 and D620. We thus propose that the R532:D620 bond that mediates the S5/

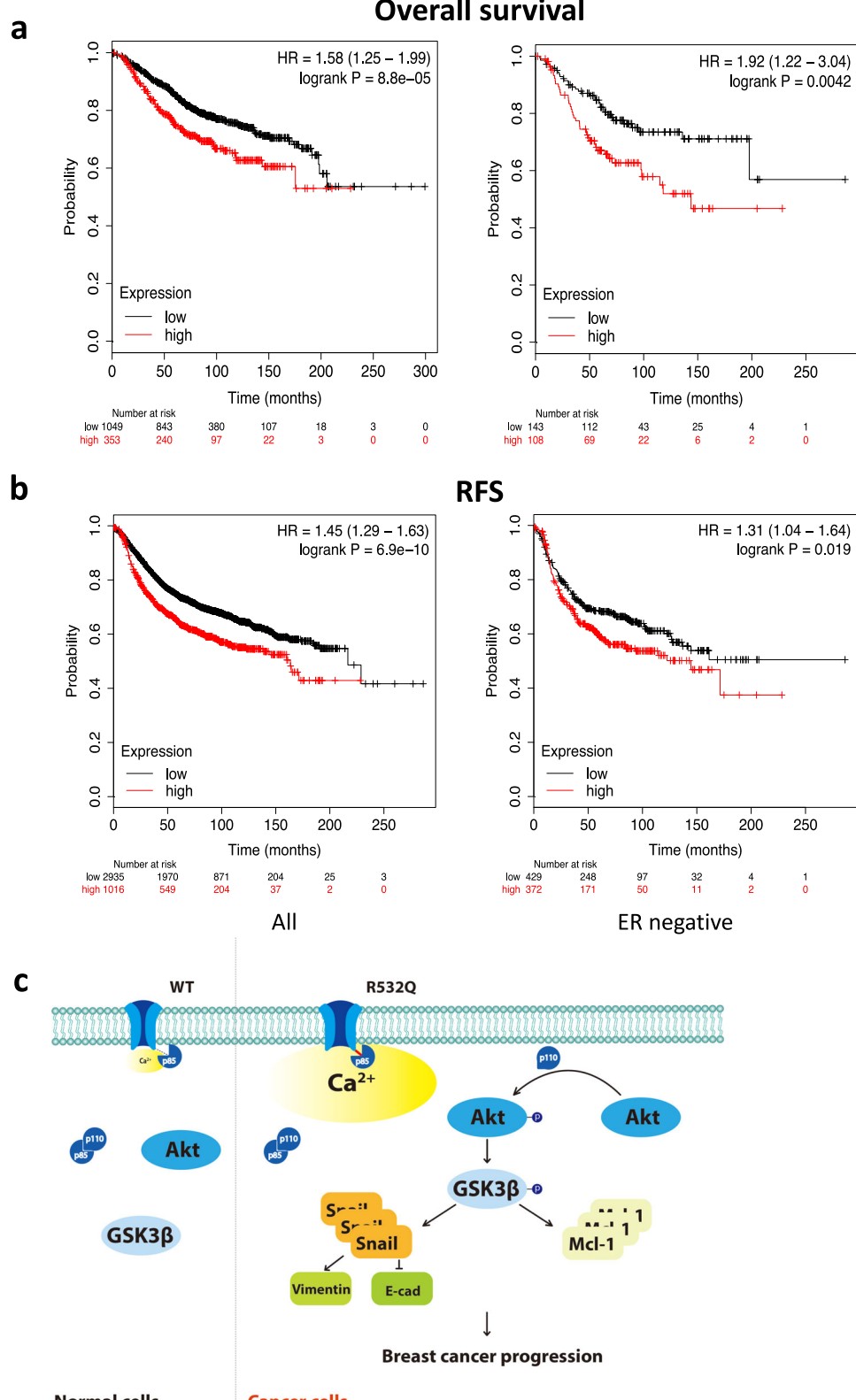

**Fig. 7 Correlation between the TRPV6 gene expression and survival in breast cancer patients.** Overall (**a**) and relapse-free survival (RFS). (**b**) of all types (left) or ER− (right) breast cancer patients analyzed by Kaplan–Meier plots. Hazard ratios (HR) and *p* values are indicated. **c** Diagram showing the hypothetical mechanism of how the R532Q promotes breast cancer cell progression through a PI3K/Akt/GSK-3β pathway.

S6 helix interaction acts as an autoinhibitory switch to control the TRPV6 channel function under physiological conditions.

Based on sequence alignments, residue R532 in helix S5 and D620 in S6 are highly conserved across both the species and TRPV members (Fig. S2), suggesting their potential importance for the TRPV subfamily during evolution. Further, despite of modest overall sequence homology among the 28 TRP members, the conservation of the two residues is also found in the other TRP subfamilies and across the species (Fig. S2). While TRPV6 is the first TRP channel known to be regulated by an autoinhibitory S5/S6 helix interaction it remains to be determined what are the functional roles of the two residues in other TRP channels. Interestingly, voltage-gated K channel EXP2, that possesses similar structural arrangements as TRPs, is also functionally regulated by G421 in helix S5 and C480 in S6, as substitution with bulky residues (Trp, Tyr, or Phe) constitutively opens the channel[28], suggesting possible presence of hydrogen bonding at G421:C480 that mediates the S5/S6 helix interaction.

According to the cancer database COSMIC there are so far 565 coding mutations in the *TRPV6* gene, which were identified in various tissues such as breast, intestine, liver, lung, skin, stomach, and urinary tract[17]. Nonetheless, the functional importance of these mutations is mostly undetermined. The GOF mutation R532Q characterized in the present study was previously predicted to be pathogenic and found in endometrium and skin, but without further reported characterization. HEK293 cells transiently expressing mutant D620V have elevated cytosolic Ca, suggesting its GOF nature but no link to the R532:D620 bonding was mentioned[26]. Recessive mutations located in the TRPV6 S2, S3, cytosolic S2–S3 loop, and N-terminus are associated with transient neonatal hyperparathyroidism characterized by impaired maternal-fatal Ca transport in placenta and unstable TRPV6 protein[29]. The TRPV6 channel function is also known to be rate-limiting for dietary Ca absorption in intestine and be important for Ca stone formation in kidney. Triple mutation C157R/M378V/M681T in TRPV6 is more likely in patients with kidney stone than in healthy people[30], and is possibly a GOF mutation based on functional analysis although the underlying mechanism remains unclear.

Elevated TRPV6 mRNA expression was first found in prostate cancer patients[31,32] and was then also observed in breast, colon, ovary, and thyroid cancers. In prostate cancer, elevated TRPV6 expression was found to be regulated through a Ca/nuclear factor of activated T-cell transcription factor (NFAT) or an Orai1/Annexin/S100A11 pathway[33,34] but how the expression is upregulated in breast cancer is poorly understood[10]. In breast cancer cell line T-47D ER antagonist tamoxifen inhibits TRPV6 channel function thereby impairing cell viability[35], but subsequent studies using TRPV6-transfected MCF-7 and MDA-MB231 cells found that the effect of tamoxifen is ER independent[18]. TRPV6 expression is essential for malignant migration and invasion of MDA-MB-231 cells[36]. Similar oncogenic roles of elevated TRPV6 in prostate, ovarian, thyroid, and pancreatic cancers have been reported[37]. In contrast, overexpression of TRPV6 inhibited colon cancer cell progression[38], which is consistent with a recent study using zebrafish epithelia and human colon carcinoma cells and showing that Trpv6 affects phosphatase 2 A thereby dephosphorylating Akt and resulting in cell quiescence[39]. Therefore, the effects of TRPV6 may be stimulating or inhibiting depending on the cancer cell type, which can be accomplished through distinct signalling factors or pathways. In summary, the involvement of a PI3K/Akt/GSK-3β pathway and TRPV6/p85 interaction in breast cancer cell progression revealed in the present study not only confirms the previously reported role of Ca, but more importantly, provides a mechanistic understanding of the disease

development. Future studies would determine the role of TRPV6 with an associated regulatory pathway in other cancers.

## Methods

**Plasmids, reagents, and antibodies.** *Xenopus* oocyte expression vector pBSMXT-MCS encoding human TRPV6 (accession number: Q9H1D0, containing 40 additional amino-acid residues at the start of the N-terminus compared to the one used in structural studies) cDNA was generated previously[6]. For mammalian cell expression, human TRPV6 cDNA was subcloned into a pCMV vector. Q5 hot start high-fidelity master mix (New England Biolabs) was used to introduce mutation(s), which were verified by sequencing. TRPV6 antibody was generated as described previously[40]. Commercial antibodies against β-actin, Mcl-1, and p85 were from Santa Cruz Biotechnology (Santa Cruz, CA) and those against p-Akt, Akt, p-GSK-3β, GSK-3β, Snail, e-cadherin, vimentin, GST, His, and HA-tag were from Cell Signaling Technology (Danvers, MA), with 1:1000 dilution for WB. DTT and MONNA were purchased from Fisher Scientific (Ottawa, ON, Canada) and Tocris Bioscience (Bristol, UK), respectively. Menthol and crystal violet were from Sigma–Aldrich (St. Louis, MO).

**Protein expression in oocytes.** WT or mutant plasmids were linearized and capped RNAs were in vitro synthesized using mMESSAGE mMACHINE T3 transcription kit (Invitrogen, Waltham, MA). Same as previously described, 25 ng cRNA was injected into each oocyte[41]. Control oocytes were those injected with water. After maintenance for 2 days, injected oocytes were subjected for WB experiments. The present study was reviewed and approved by Ethical Committee for Animal Experiments of the University of Alberta and conducted following the Guidelines for Research with Experimental Animals of the University of Alberta and the Guide for the Care and Use of Laboratory Animals (NIH Guide) revised in 1996.

**Two-microelectrode voltage clamp in oocytes.** Microelectrodes made from capillary pipettes were cut with resistance in the 0.3–2 MΩ range and filled with 3 M KCl. An oocyte was penetrated by two electrodes and whole-oocyte currents were measured with perfusion of extracellular solutions. Currents and voltages were obtained using a Geneclamp 500B amplifier and Digidata 1322 A AD/DA converter (Molecular Devices, Union City, CA) at 200 μs/sample and Bessel filtered at 2 kHz. The files were analyzed with pClamp 11 (Axon Instruments, Union City, CA) and data plotted with GraphPad Prism 8 (GraphPad Software, San Diego, CA).

**Western blotting and biotinylation.** Oocytes and mammalian cells subjected to WB experiments were all lysed in CelLytic M lysis buffer (Sigma) with proteinase inhibitor cocktail (Thermo Scientific, Waltham, MA). Harvested proteins were separated by SDS-PAGE and transferred to PVDF membrane. After slim milk blocking at room temperature, a membrane was incubated with diluted antibodies overnight at 4 °C. Secondary antibodies conjugated with HRP were purchased from GE healthcare. Biotinylation assays were performed as described previously[6]. Oocytes were washed with ice-cold PBS and incubated with 0.5 mg/ml sulfo-NHS-SS-Biotin (Pierce, Rockford, IL), followed by quenching 1 M NH₄Cl. Surface proteins bound to streptavidin was subjected to WB procedures.

**Immunoprecipitation and TEV cleavage.** Oocytes expressing TRPV6 mutant R532C/D620C_TEV were lysed and incubated with TRPV6 antibody plus 50% protein G slurry overnight at 4 °C. 0.6 mM glutathione mixed with 0.4 mM oxidized glutathione or 1 mM DTT along with AcTEV protease (Invitrogen) was added to the mixtures[42]. Cells with stable expression of TRPV6 or a mutant were lysed and incubated with TRPV6 antibody plus 50% protein G sepharose (GE Healthcare) overnight at 4 °C. The proteins were eluted with SDS loading buffer and analyzed by WB.

**Immunofluorescence.** Oocytes or mammalian cells were washed with ice-cooled PBS and fixed with 4% paraformaldehyde, followed by permeabilization with Triton. After blocking with 3% skim milk or BSA, oocytes or mammalian cells were incubated with specific antibodies overnight at 4 °C and then with Alexa fluor 488 anti-rabbit IgG (Invitrogen) at room temperature for 1 hour (hr). Fisher Chemica Permount Mounting medium (Fisher Scientific) was used to mount the oocytes, while ProLong Gold Antifade Mountant (Molecular Probes, Life Technologies, Grand Island, NY) was used to mount mammalian cells. Fluorescence was examined with an Olympus IX-81 spinning confocal microscope (Cell Imaging Centre, Faculty of Medicine and Dentistry, University of Alberta).

**In vitro pull-down.** The full N-terminus, S5 and S6 fragments of TRPV6 were amplified and subcloned into vectors pET-28a and pGEX-5X and pMBP-His as a gift of Dr. Joanne Lemieux (University of Alberta)[43]. *E. coli.* expression plasmids encoding nSH2, SH3, and BH of p85 were gifts of Dr. Lewis Cantley (Cornell University) and one encoding cSH2 was from Dr. Sharona E. Gordon (University

of Washington). Peptides with His or GST tag produced from *E. coli.* were purified with corresponding GST (GE Healthcare, Waukesha, WI) or Ni-NTA beads (Qiagen, Hilden, Germany). Purified peptides were incubated together overnight at 4 °C and supplemented with 50% beads. After extensively wash with 1% NP-40, beads conjugated with peptides were denatured and subjected WB analysis. Uncropped WB images were included in Fig. S15.

**MD simulation**. The initial simulation system was set up based on the cryo-EM structure of human TRPV6 (PDB: 6D7T)[44]. Only the pore region of TRPV6 (G511-T640) was used for simulation. This region contains the transmembrane helix S5, pore helix, S6, and TRP helix. The Y507A mutation and 2-APB in the initial 6D7T structure were absent during the simulation. The TRPV6 pore structure was then modeled in a 1-palmitoyl-2-oleoyl-sn-glycero-3-phosphocholine (POPC) lipid bilayer using CHARMM-GUI membrane builder[45] (Fig. S8a and b). The system was hydrated with a total of 12,950 TIP3P water molecules on both sides of the bilayer. The approximate dimension of the resultant simulation box was $87 \times 87 \times 99$ Å along the x, y, and z axes, respectively. $Na^+$ and $Cl^-$ ions were added to the system for neutralization and maintain a concentration of 150 mM NaCl. The two $Ca^{2+}$ ions in the PDB structure were retained in the simulation system. The input parameters for the protein, ions, water molecules and lipids were generated by CHARMM-GUI Input Generator[46]. To investigate the effect of the R532Q mutation on the structural and dynamic changes in TRPV6, the four residues R532 in the tetrameric TRPV6 were mutated to Q532 using the muta-genesis function of PyMOL. The two systems are noted as WT TRPV6 and TRPV6 R532Q.

MD simulation was carried out using the AMBER18 simulation package similar to one previously described[47]. Simulation for a total of 400 ns was performed (Fig. S8c). Before data analysis, the root mean square deviation (RMSD) for the Cα atoms of TRPV6 was calculated to assess the equilibration of the simulation. The RMSD results showed that the two simulations reached equilibrium after 200 ns. Thus, the trajectories of the last 200 ns simulation were used for data analyses. All the simulation analyses were performed using the CPPTRAJ program of AMBER18. The VMD software was used for structure visualization[48]. A hydrogen bond was identified if the distance between the acceptor and donor heavy atoms is less than 3.0 Å and the angle formed by the acceptor heavy atom/donor hydrogen atom line and the donor hydrogen atom/donor heavy atom line is greater than 135°.

**Cell culture and ratiometric Ca imaging**. MCF-7 and MDA-MB-231 cells were gifts of Dr. Zhixiang Wang (University of Alberta) and were cultured in RPMI1640 or DMEM supplied with 10% FBS (Gibco) and 1% penicillin streptomycin as the standard culture medium. Stable expression of WT TRPV6 and mutant R532Q in both cell lines was generated through transfection with Fugene HD (Promega, WI) and selected with G418 (Gibco). 300 μg/ml G418 was added for maintenance of stable cell lines. Ca imaging was performed as described previously[6]. Stable cells plated on glass coverslips were loaded with culture medium added with 5 μM Fura-2-AM in the dark and incubated at 37 °C for 30 min. Cells were then washed with bath solution (in mM): 1 MgCl2, 2 CaCl2, 4 KCl, 140 NaCl, 10 HEPES, 10 glucose, with pH adjusted to 7.2 with NaOH. CaCl2 was replaced by MgCl2 to form a nominal Ca-free solution. A Leica DMI6000B microscope with Quorum MAC6000 modular automation controller system and a Lambda DG4P-215 lamp were used for all measurements. Fura-2 was alternately excited at 340 nm and 380 nm for 30 ms every 2 s and the emitted fluorescence was recorded with a Hamamatso Orca-Flash 4.0 digital camera. F340 and F380 pictures were used to calculate the ratiometric images after background correction, i.e., subtraction by the fluorescence intensity in a cell-free area corresponding to 340 and 380 nm excitation, respectively. Single cells were marked as regions of interest and the F340/F380 ratio was plotted versus time.

**Cell proliferation assay**. $5 \times 10^3$ stably transfected cells were seeded in a 96-well plate with triplicate for each group with the standard culture medium. Cell counting kit-8 (Sigma–Aldrich) was used according to manufacturer's instruction. After incubation for 4 h, OD values were measured with a microplate reader.

**In vitro scratch assay**. To halt cell proliferation, mitomycin (Sigma–Aldrich) was added 2 h before experiments. After a monolayer of cells was formed, scratches were made (0 h) with a sterilized fine tip. Images were taken by a digital camera with C-Mount on randomly selected areas for each group. At 24 h (MCF-7 cells) or 16 h (MDA-MB-231 cells) images of the same selected areas were taken. The standard culture medium was used throughout.

**Invasion assay**. Invasion assays were performed using costar transwell with 8.0 μm pore polycarbonate inserts and with the use of growth factor reduced Corning Matrigel invasion chamber (Bedford, MA). $5 \times 10^3$ stably transfected MDA-MB-231 cells were seeded on the upper chamber with starvation medium without FBS overnight. Culture medium supplemented with 20% FBS was added to the lower chamber. Twelve hours later, the assays were stopped and cells remained in the upper chamber were removed with cotton swabs. The invaded cells on the inserts were fixed with menthol and stained with crystal violet.

**Soft agar colony formation assay**. 2500 stably transfected cells were mixed with 0.35% low temperature gelling agarose (Sigma–Aldrich) and added to the 0.7% base soft agar in a 24-well plate. Cells were fed twice per week. Three weeks later, cells were imaged and the number of colonies formed counted and analyzed.

**Correlation of the TRPV6 expression with breast cancer patient survival**. The relevance of the TRPV6 expression to the survival rate in breast cancer patients was examined using published databases available through the Kaplan–Meier plotter[24]. The curves of overall survival and relapse-free survival, with the logrank $P$-value and hazard ratio with 95% confidence interval, were generated by segregating patients with high or low expression of the *TRPV6* gene (206827_s_at).

**Statistics and reproducibility**. Data presented here were expressed as mean ± SEM (standard error of the mean). *, **, and *** were assigned when the *p* values calculated by Student's *t*-test for two groups comparisons and one-way ANOVA for multiple groups comparisons were less than 0.05, 0.01, and 0.001, respectively. Statistically not significant was abbreviated as 'ns'. N numbers are included in corresponding figure legends. All experiments are repeated three times.

**Reporting summary**. Further information on research design is available in the Nature Research Reporting Summary linked to this article.

## Data availability
All source data are available in Supplementary Data 1. MD simulation data are available upon reasonable request to J.-B.P.

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

## Acknowledgements

This work was supported by the Natural Sciences and Engineering Research Council of Canada (NSERC), the Kidney Foundation of Canada (to X.Z.C.), and the National Natural Science Foundation of China (grant # 31871176 and 32070726, to J.T.).

## Author contributions

Conceptualization, R.C. and X.-Z.C.; Investigation, R.C., L.-Y.W. and X.L. Supervision, J.T., J.-B.P. and X.-Z.C. Writing, R.C., L.-Y.W., J.-B.P. and X.-Z.C.

## Competing interests

R.C. was a recipient of the Dean's Doctoral Awards, Alberta Innovates Graduate Student Scholarship, and the NSERC International Research Training Group Studentship. The remaining authors declare no competing interests.
