## [Transparent Peer Review File · Communications Biology]

Reviewers' comments:

Reviewer #1 (Remarks to the Author):

Cai and colleagues identify a functionally important intramolecular interaction in the Ca²⁺ selective ion channel TRPV6. Their functional data suggests that the R532 residue in S5 and the D620 residue in S6 interact in the context of the intact channel, and this interaction stabilizes the closed state of the channel. The Catalogue of Somatic Mutations in Cancer database predicts that the R532Q mutation is pathogenic in cancer, which gives special significance to the findings. Charge neutralization, or charge reversal mutations of both residues dramatically increase Ca²⁺ currents without substantial effects on surface expression. Charge swapping of the two residues on the other hand resulted in phenotypes similar to wild type channels. Replacing these two residues with cysteines resulted in channels with very small current, which were increased substantially by DTT, as opposed to wild-type channels which were not affected by DTT. After introducing a protease site in the double cysteine mutant, a smaller fragment showed up on the gel only in DTT treated cells, indicating that the disulfide bond held the channel together. Pulldown experiments also confirmed the roles of the R532 and D620 residues in interaction between the S5 and S6 segment. Molecular dynamics simulations indicated that in the initial configuration there was no hydrogen bond between the two residues, but they developed during the 400 ns simulations. Ca²⁺ imaging experiments confirmed the high basal activity of the R532Q mutation, and breast cancer cell lines expressing the mutant proliferated faster than cells expressing the wild-type channel. The authors also show increased Akt phosphorylation in cells overexpressing the wild type and mutant channels, and they demonstrate interaction of the channel with endogenous p85 in a Ca²⁺ dependent manner, and this interaction was increased by the R532Q mutation. Finally they show using published breast cancer databases that the TRPV6 expression correlates with poor survival.

The authors present a compelling set of data, with a reasonable interpretation on an important ion channel, and show relevance for cancer progression. In this reviewer's opinion the manuscript will have significant impact on the field, thus it is well worth publishing. I have a number of comments to improve the presentation, as well as better understand the mechanism of overactivity of these mutants.

1. Are monovalent currents also increased by these mutations? This is an important question, as steady state Ca²⁺ currents are a result of a balance between channel opening and Ca²⁺ induced inactivation, which was shown to be mediated by calmodulin binding to the channel as well as PIP2 depletion. Both residues are very close to the pore, where CaM inserts itself to block the channel, and thus can affect CaM inhibition. They can also affect open state stability of the channel, in that case monovalent currents should also increase.
2. Figure 1 Western blots. The authors claim that there is no difference between surface expression of the different mutants. The representative images show some differences in intensity, even 2-3 fold by eye. Even though it is nowhere near the over 50 fold increase in currents, quantification would be advisable.
3. Figure 2B: the gel should be quantified.
4. Figure 5D. Please provide details on the BAPTA-AM treatment of the cells. What concentration was used, how long, and how much before the experiment? Please also disclose the number of experiments, and provide summary with densitometry as in several other figures. Also, intracellular BAPTA-AM may be overridden by the large amount of Ca²⁺ entering the cell through the mutant. The authors may consider repeating the experiment in the absence of extracellular Ca²⁺.
5. Supplemental Figure 3C: What does the red trace show and what does the black trace show?

6. The authors should provide more details on H bond formation was predicted computationally.

7. What was the reason the authors used a truncated TRPV6, consisting only S5 and S6 for MD simulations.

Minor:

1. Page 4. Line 85: PI3K can be recruited to near the surface membrane and activated by receptors or adaptor molecules (12) which could include TRPV6. This can be read as if there is data on TRPV6 serving as adapter for PI3K, which I believe is not the case. The authors may consider modifying this sentence

2. Page 8, line 169: "To test whether it is the case during the simulation, a hydrogen bond between a residue (R or Q) at 532 and D620 was calculated." This sentence does not sound grammatically correct. Please modify.

3. Page 15 line 354: "The TRPV6 channel function is also known to be rate limiting for dietary Ca absorption in intestine and Ca stone formation in kidney." The second part of the sentence sounds odd. TRPV6 is rate limiting for dietary Ca²⁺ absorption, but it is unclear what it means to be rate limiting for kidney stone formation. Also, my understanding is that people with the potentially overactive triple mutant develop kidney stones, because they absorb more Ca²⁺ in the intestines, and the kidney compensates by excreting more Ca²⁺ which increases the chance of stone formation.

4. The numbering of the R532 and D620 residues are based on the relatively recently identified longer isoform of TRPV6 with 40 amino acids added to the N-terminus. While this may be the physiological isoform, most data in the literature, including all structures use the old numbering where R532 is R492, whereas D620 is D580. I would recommend clarifying this in the MS somewhere, so that people outside the narrow field do not get confused with the numbers.

5. Fig 2A: DTT and control are labeled with closed circles and closed squares. The resolution of the figure is so poor, that it is very hard to see what is a circle what is a square. Please fix. Same applies to Fig. 5B and C.

6. Fig. 4 B & C. The Ca²⁺ traces should be colored. In panel C is somewhat ambiguous which trace is which where Ca²⁺ is added

Reviewer #2 (Remarks to the Author):

The Ca-selective TRPV6 channel is predominantly expressed in epithelial cells and overexpression of the TRPV6 gene was proposed to be associated with the malignancy of epithelial tumours by several groups, including the group of the authors of this manuscript.

Here the authors investigated the effects of the potentially pathological TRPV6R532Q mutation - based on data from the COSMIC database - on the TRPV6 current in the oocyte expression system and on several in vitro cell-based assays (Ca-imaging, cell proliferation and migration) in breast cancer cell lines MCF-7 and MDA-MB-231. The results indicate a continuous contact through which the 532R residue of the transmembrane segment 5 of the channel is in contact with D620 at the distal end of the S6 segment which traps the lower gate of the TRPV6 channel in a low ion conducting conformation. The R532Q mutation disrupts this 532R-D620 contact yielding a gain-of-function phenotype considering ion current and cytosolic Ca and changes of cell proliferation and migration. The 532R-D620 contact is further supported by pull-down assays, molecular dynamics simulation and the DTT-sensitive disulfide bond formation of a R532C/D620C double mutant. Furthermore, PI3 kinase Akt signalling is activated and the authors show that the TRPV6 N-terminus interacts with the p85

subunit of PI3 kinase.

The data are of interest to a broad audience, mostly because data place TRPV6 into the centre of cancer pathologies which is relevant for physiology and pathophysiology in humans.

The experiments are well done. In the methods section there is a short paragraph on "statistical analysis" which is not applicable to all calculations in the manuscript with $n > 2$ groups. It is more appropriate to state the statistical analysis in the figure legends. The level of detail provided for others to reproduce the work could be improved.

- 1) Please adjust the title to match the data shown in the manuscript, changes in migration and proliferation of cells transfected with TRPV6 wild-type and mutant cDNAs: I recommend a less dogmatic conclusion.
- 2) If available, the authors could show pull-down controls (Fig. 2C) with GST-S5 and HIS-MBP only, and with HIS-MBP-S6 and GST only, to rule out non-TRPV6 interactions.
- 3) The TEV experiment (Fig. 2B) should be explained: Where does the 24 kDa fragment come from?
- 4) Is the ion selectivity altered by the introduced mutations R532Q and D620 R?
- 5) Culture conditions for TRPV6 wild-type and mutant cDNAs expressing MCF-7 and MDA-MB-231 cells. According to Fig. 4B and C, basal Ca is significantly elevated in cells transfected with mutant TRPV6 even in the absence of extracellular calcium, suggesting toxic Ca effects during cell culture: Culture medium usually contains calcium at concentrations of 1.5 to 2mM. The authors should elaborate on this point.
- 6) Please change "wound healing assay" and "wound closure assay" to "in vitro scratch assay".
- 7) Cell proliferation and migration is highly dependent on the serum concentration in the medium: the respective serum concentration should be stated in the methods section.
- 8) Figs. 6 and 7C: It has been shown that the PI3K pathway can be activated by blocking TRPV6 but is not activated in the absence of blockade (PMID: 31526479). Authors should elaborate on this point in the Discussion and add the reference.

Minor:

Lanes 448 and 690: MD simulation instead of "stimulation"

Lane 487: Mytomycin or Mitomycin?

Accession number of human TRPV6 should be included (765 aa?)

Cosmic data base: In which samples/cancerous tissues was the TRPV6R532Q mutation identified? (any reference?)

Reviewer #3 (Remarks to the Author):

In this manuscript, Cai et al. study the role of a specific site of interaction between S5 and S6 helices of the TRPV6 channel. The authors used a variety of techniques to demonstrate the existence and physiological importance of this interaction. The authors found that a hydrogen bond between R532 and D620 is important for the channel's gating process. Moreover, a mutation in position R532 (R532Q) was previously shown to occur in breast cancer (data from a public site). Thus, this study's results shed light on the gating process of TRPV6 and its physiological/pathophysiological role. The manuscript is well written and relatively easy to follow. However, in my opinion, several aspects of the study should be clarified before this manuscript will be ready for publication.

1. The authors show that the R532Q mutation results in dramatic GOF phenomena (50 times increase of the basal current). The authors suggest that this result from the increase in the pore diameter based on their MD analysis in the basal state of the channel. However, the authors do not provide any evidence that the channel gating remains intact following the mutations. I would suggest including an

experiment testing the mutate channel response to allosteric inhibitors (e.g., 2-APB).

2. The author's conclusions regarding the increase in the pore diameter due to the mutation can be proven by performing a single-channel analysis. However, due to the technical issues regarding this kind of experiment, the authors should perform it only if it is a common technique in their lab.

3. Although my expertise is more into the biophysical aspects of ion channels, I found the cancer part of the manuscript quite puzzling. The authors transfect the wild-type and mutated TRPV6 channels into cancerous cells that do not express the channel at all. What is the point of such an experiment? Any GOF mutation in a calcium-permeable channel will result in similar results. Why do the authors not show the difference in cancerous cells that natively express TRPV6?

Responses to reviewers' comments (in bold and start with ">>"; others are referees' original comments)

Reviewers' comments:

Reviewer #1 (Remarks to the Author):

Cai and colleagues identify a functionally important intramolecular interaction in the Ca²⁺ selective ion channel TRPV6. Their functional data suggests that the R532 residue in S5 and the D620 residue in S6 interact in the context of the intact channel, and this interaction stabilizes the closed state of the channel. The Catalogue of Somatic Mutations in Cancer database predicts that the R532Q mutation is pathogenic in cancer, which gives special significance to the findings. Charge neutralization, or charge reversal mutations of both residues dramatically increase Ca²⁺ currents without substantial effects on surface expression. Charge swapping of the two residues on the other hand resulted in phenotypes similar to wild type channels. Replacing these two residues with cysteines resulted in channels with very small current, which were increased substantially by DTT, as opposed to wild-type channels which were not affected by DTT. After introducing a protease site in the double cysteine mutant, a smaller fragment showed up on the gel only in DTT treated cells, indicating that the disulfide bond held the channel together. Pulldown experiments also confirmed the roles of the R532 and D620 residues in interaction between the S5 and S6 segment. Molecular dynamics simulations indicated that in the initial configuration there was no hydrogen bond between the two residues, but they developed during the 400 ns simulations. Ca²⁺ imaging experiments confirmed the high basal activity of the R532Q mutation, and breast cancer cell lines expressing the mutant proliferated faster than cells expressing the wild-type channel. The authors also show increased Akt phosphorylation in cells overexpressing the wild type and mutant channels, and they demonstrate interaction of the channel with endogenous p85 in a Ca²⁺ dependent manner, and this interaction was increased by the R532Q mutation. Finally they show using published breast cancer databases that the TRPV6 expression correlates with poor survival.

The authors present a compelling set of data, with a reasonable interpretation on an important ion channel, and show relevance for cancer progression. In this reviewer's opinion the manuscript will have significant impact on the field, thus it is well worth publishing. I have a

number of comments to improve the presentation, as well as better understand the mechanism of overactivity of these mutants.

>> We would like to thank the reviewer for the very positive appraisal and enthusiasm about our study. Please find our point-by-point responses below.

1. Are monovalent currents also increased by these mutations ? This is an important question, as steady state Ca^{2+} currents are a result of a balance between channel opening and Ca^{2+} induced inactivation, which was shown to be mediated by calmodulin binding to the channel as well as PIP2 depletion. Both residues are very close to the pore, where CaM inserts itself to block the channel, and thus can affect CaM inhibition. They can also affect open state stability of the channel, in that case monovalent currents should also increase.

>> This is an excellent point. We have now examined Na currents mediated by mutants R532Q and D620A and found that they are significantly increased compared with WT TRPV6-mediated Na currents (see Fig. S4 in the revised manuscript).

2. Figure 1 Western blots. The authors claim that there is no difference between surface expression of the different mutants. The representative images show some differences in intensity, even 2-3 fold by eye. Even though it is nowhere near the over 50 fold increase in currents, quantification would be advisable.

>> We agree with the reviewer observation on the representative images. However, our statistical analysis showed that these mutations do not significantly alter the surface expression (see Fig. S1 and S3).

3. Figure 2B: the gel should be quantified.

>> Done (Fig. S6).

4. Figure 5D. Please provide details on the BAPTA-AM treatment of the cells. What concentration was used, how long, and how much before the experiment? Please also disclose the number of experiments, and provide summary with densitometry as in several other figures. Also, intracellular BAPTA-AM may be overridden by the large amount of Ca^{2+} entering the cell through the mutant. The authors may consider repeating the experiment in the absence of extracellular Ca^{2+} .

>> We included the details in the corresponding figure legend (page 34, line 776). All experiments were performed three times independently. Gel quantification for this and other applicable figures has been included as Fig. S1, S3, S6, S9, S10, S11, S12 and S13. To investigate the role of extracellular Ca, we incubated the cells with 2 mM EGTA for 1 hr and found similar results as using membrane permeable BAPTA-AM (Fig. S10), suggesting that BAPTA-AM effectively chelated intracellular Ca in the presence of extracellular Ca.

5. Supplemental Figure 3C: What does the red trace show and what does the black trace show?

>> The red and black traces in Figure 3C show the RMSD values of TRPV5 with and without the R532Q mutation, respectively. We have now added “The red and black lines show the RMSD values of TRPV6 with and without R532Q mutation, respectively, during the 400 ns simulations” in the revised figure legend.

6. The authors should provide more details on H bond formation was predicted computationally.

>> The hydrogen bond analysis was performed by the CPPTRAJ program of AMBER18 simulation package. A hydrogen bond was identified if the distance between the acceptor and donor heavy atoms is less than 3.0 Å and the angle formed by the acceptor heavy atom/donor hydrogen atom line and the donor hydrogen atom/donor heavy atom line is greater than 135°. We have now added these details in Methods (page 19, line 492).

7. What was the reason the authors used a truncated TRPV6, consisting only S5 and S6 for MD simulations.

>> Our MD simulations only focused on the TRPV6 pore region, consisting of S5, pore helix and S6, which together define the selectivity filter and the (lower) pore gate, would be sensitive to the R532Q mutation, and thus are central to the TRPV6 function. We also wanted to make sure that our calculations can be completed within a reasonable period of time using the computational resource at the Alabama Supercomputer Center. It would take at least one year to simulate the full-length TRPV6. Note that in one of our previous MD studies, simulations on this region of TRPV5, a close homologue of TRPV6, have provided functional insights into the dynamic changes caused by the A563T mutation (Wang et al., Biochemistry, 2016. PMID: 26837804).

Minor:

1. Page 4. Line 85: PI3K can be recruited to near the surface membrane and activated by receptors or adaptor molecules (12) which could include TRPV6. This can be read as if there is data on TRPV6 serving as adapter for PI3K, which I believe is not the case. The authors may consider modifying this sentence.

>> The part “which could include TRPV6” in the sentence has now been removed.

2. Page 8, line 169: To test whether it is the case during the simulation, a hydrogen bond between a residue (R or Q) at 532 and D620 was calculated. This sentence does not sound grammatically correct. Please modify.

>> This sentence is now changed to “To test whether it is the case during the simulation,

the formation of a hydrogen bond between residues R532 (or Q532) and D620 was calculated”.

3. Page 15 line 354: "The TRPV6 channel function is also known to be rate limiting for dietary Ca absorption in intestine and Ca stone formation in kidney." The second part of the sentence sounds odd. TRPV6 is rate limiting for dietary Ca²⁺ absorption, but it is unclear what it means to be rate limiting for kidney stone formation. Also, my understanding is that people with the potentially overactive triple mutant develop kidney stones, because they absorb more Ca²⁺ in the intestines, and the kidney compensates by excreting more Ca²⁺ which increases the chance of stone formation.

>> This sentence is now changed to “The TRPV6 channel function is also known to be rate limiting for dietary Ca absorption in intestine and be important for Ca stone formation in kidney”.

4. The numbering of the R532 and D620 residues are based on the relatively recently identified longer isoform of TRPV6 with 40 amino acids added to the N-terminus. While this may be the physiological isoform, most data in the literature, including all structures use the old numbering where R532 is R492, whereas D620 is D580. I would recommend clarifying this in the MS somewhere, so that people outside the narrow field do not get confused with the numbers.

>> Thanks for this important point. We have now included this information as well as the accession number in Methods (page 16, line 394).

5. Fig 2A: DTT and control are labeled with closed circles and closed squares. The resolution of the figure is so poor, that it is very hard to see what is a circle what is a square. Please fix. Same applies to Fig. 5B and C.

>> These figures have now been fixed according to the comment.

6. Fig. 4 B & C. The Ca²⁺ traces should be colored. In panel C is somewhat ambiguous which trace is which where Ca²⁺ is added.

>> The traces are now in blue and red for WT- and R532Q-expressing cells, respectively (Fig. 4B and C).

Reviewer #2 (Remarks to the Author):

The Ca-selective TRPV6 channel is predominantly expressed in epithelial cells and overexpression of the TRPV6 gene was proposed to be associated with the malignancy of epithelial tumours by several groups, including the group of the authors of this manuscript. Here the authors investigated the effects of the potentially pathological TRPV6R532Q mutation - based on data from the COSMIC database - on the TRPV6 current in the oocyte expression

system and on several in vitro cell-based assays (Ca-imaging, cell proliferation and migration) in breast cancer cell lines MCF-7 and MDA-MB-231. The results indicate a continuous contact through which the 532R residue of the transmembrane segment 5 of the channel is in contact with D620 at the distal end of the S6 segment which traps the lower gate of the TRPV6 channel in a low ion conducting conformation. The R532Q mutation disrupts this 532R-D620 contact yielding a gain-of-function phenotype considering ion current and cytosolic Ca and changes of cell proliferation and migration. The 532R-D620 contact is further supported by pull-down assays, molecular dynamics simulation and the DTT-sensitive disulfide bond formation of a R532C/D620C double mutant. Furthermore, PI3 kinase Akt signalling is activated and the authors show that the TRPV6 N-terminus interacts with the p85 subunit of PI3 kinase.

The data are of interest to a broad audience, mostly because data place TRPV6 into the centre of cancer pathologies which is relevant for physiology and pathophysiology in humans.

The experiments are well done. In the methods section there is a short paragraph on "statistical analysis" which is not applicable to all calculations in the manuscript with $n > 2$ groups. It is more appropriate to state the statistical analysis in the figure legends. The level of detail provided for others to reproduce the work could be improved.

>> We would like to thank the reviewer for the very positive appraisal of our study. We have now included statistical analysis in figure legends where applicable.

1) Please adjust the title to match the data shown in the manuscript, changes in migration and proliferation of cells transfected with TRPV6 wild-type and mutant cDNAs: I recommend a less dogmatic conclusion.

>> We agreed and have now changed from "breast cancer progression" to "breast cancer cell progression" in the manuscript title and throughout the manuscript. We used 'progression' as a summarizing word for proliferation, migration, invasion, and colony formation of breast cancer cell lines.

2) If available, the authors could show pull-down controls (Fig. 2C) with GST-S5 and HIS-MBP only, and with HIS-MBP-S6 and GST only, to rule out non-TRPV6 interactions.

>> Yes, these control data have now been included (Fig. S7).

3) The TEV experiment (Fig. 2B) should be explained: Where does the 24 kDa fragment come from?

>> Our TRPV6 antibody recognizes its C-terminus. With the insertion of the TEV cleavage site, TEV should cleave TRPV6 into two fragments, of which the smaller one (~24 kD) contains the C-terminus recognized by the antibody. However, in the absence of DTT there was basically no detectable 24 kD band because the two fragments stay together by the disulfide bond formed between C532 and C620 (Fig. 2B). This has now been explained in the revised manuscript (page 8, line 161).

4) Is the ion selectivity altered by the introduced mutations R532Q and D620R?

>> We evaluated the Ca to Na permeability ratio P_{Ca}/P_{Na} , which depends on the difference between their reversal potentials ($E_{rev}(Ca) - E_{rev}(Na)$). In part because the I-V curves show a significant degree of inward rectification, the $E_{rev}(Ca) - E_{rev}(Na)$ values were quite variable from one oocyte to another (see figure below). However, no significant difference in the $E_{rev}(Ca) - E_{rev}(Na)$ value was found for mutant channel R532Q or D620R compared with that for WT TRPV6 (see figure below), indicating that the two GOF mutations do not significantly alter the ion selectivity of the channel. Of note, we pre-injected oocytes with EGTA to eliminate complexity induced by uncontrolled changes in the intracellular Ca concentration due to Ca entry and obtained I-V curves in the presence of 30 mM Ca or 100 mM Na in the extracellular solution.

We propose not to include the ion selectivity in the revised manuscript. The exact Ca to Na permeability ratio could be determined if we could manipulate the intracellular ion concentrations, but our electrophysiology setting does not allow to determine these values in oocytes. Moreover, due to technical limitations and the pandemic, it'd not be feasible to timely build a corresponding setup. That being said, the ion selectivity of TRPV6 and its GOF mutants would be very interesting to investigate in the future while it's beyond our current focus. We hope the reviewer would agree.

5) Culture conditions for TRPV6 wild-type and mutant cDNAs expressing MCF-7 and MDA-MB-231 cells. According to Fig. 4B and C, basal Ca is significantly elevated in cells transfected with mutant TRPV6 even in the absence of extracellular calcium, suggesting toxic Ca effects during cell culture: Culture medium usually contains calcium at concentrations of 1.5 to 2mM. The authors should elaborate on this point.

>> We agreed with the reviewer that mutant-expressing cells should have increased Ca entry because RPMI 1640 and High glucose DMEM contain 0.42 and 1.8 mM Ca^{2+} , respectively, which should have resulted in higher basal $[Ca]_i$ levels even in the absence of extracellular Ca during measurements. We now included a sentence in Results: “The higher basal Ca levels in mutant-expressing cells may be due to increased entry of Ca contained in culture media”.

6) Please change wound healing assay and wound closure assay to in vitro scratch assay.

>> Changed throughout the manuscript.

7) Cell proliferation and migration is highly dependent on the serum concentration in the medium: the respective serum concentration should be stated in the methods section.

>> The culture medium supplemented with 10% FBS (Gibco) and 1% penicillin streptomycin is now called the standard culture medium, which is now described for proliferation and migration experiments (page 19, line 499 and page 20, lines 519, 528, 534).

8) Figs. 6 and 7C: It has been shown that the PI3K pathway can be activated by blocking TRPV6 but is not activated in the absence of blockade (PMID: 31526479). Authors should elaborate on this point in the Discussion and add the reference.

>> Excellent point. This has now been cited and elaborated in Discussion (page 15, line 381).

Minor:

Lanes 448 and 690: MD simulation instead of stimulation.

>> Done.

Lane 487: Mytomycin or Mitomycin?

>> Changed to Mitomycin.

Accession number of human TRPV6 should be included (765 aa?).

>> The accession number is now included in Methods (page 16, line 393).

Cosmic data base: In which samples/cancerous tissues was the TRPV6R532Q mutation identified? (any reference?)

>> The mutation was identified in endometrium with the sample names of TCGA-BK-A6W3-01 and TCGA-EO-A3AV-01, study name of "Uterine Corpus Endometrioid Carcinoma - TCGA, US", and study ID of COSU419 (Krauthammer et al., Nat Genet, 2012. PMID: 22842228). Besides, the mutation was also found in the skin (sample name YUWAND).

Reviewer #3 (Remarks to the Author):

In this manuscript, Cai et al. study the role of a specific site of interaction between S5 and S6 helices of the TRPV6 channel. The authors used a variety of techniques to demonstrate the

existence and physiological importance of this interaction. The authors found that a hydrogen bond between R532 and D620 is important for the channel's gating process. Moreover, a mutation in position R532 (R532Q) was previously shown to occur in breast cancer (data from a public site). Thus, this study's results shed light on the gating process of TRPV6 and its physiological/pathophysiological role. The manuscript is well written and relatively easy to follow. However, in my opinion, several aspects of the study should be clarified before this manuscript will be ready for publication.

>> We are grateful for the very positive appraisal by the reviewer.

1. The authors show that the R532Q mutation results in dramatic GOF phenomena (50 times increase of the basal current). The authors suggest that this result from the increase in the pore diameter based on their MD analysis in the basal state of the channel. However, the authors do not provide any evidence that the channel gating remains intact following the mutations. I would suggest including an experiment testing the mutate channel response to allosteric inhibitors (e.g., 2-APB).

>> We tested the effect of 5 mM 2-APB on WT TRPV6 and the R532Q and D620A mutants and found that 2-APB has comparable inhibitory effects on the mutant channels (Fig. S5). These data have now been included in Results of the revised manuscript (page 7, line 147).

2. The author's conclusions regarding the increase in the pore diameter due to the mutation can be proven by performing a single-channel analysis. However, due to the technical issues regarding this kind of experiment, the authors should perform it only if it is a common technique in their lab.

>> We would like to thank the reviewer for the understanding. Due to technical limitations and the pandemic, we found it'd be hard to conduct single channel recordings within a reasonable period of time. This would be of interest for future investigations though. We have now softened our tone in our conclusion (page 9, lines 210 to 212).

3. Although my expertise is more into the biophysical aspects of ion channels, I found the cancer part of the manuscript quite puzzling. The authors transfect the wild-type and mutated TRPV6 channels into cancerous cells that do not express the channel at all. What is the point of such an experiment? Any GOF mutation in a calcium-permeable channel will result in similar results. Why do the authors not show the difference in cancerous cells that natively express TRPV6?

>> This is reasonable and excellent question. First, we would like to clarify that TRPV6 is in fact endogenously expressed, though at relatively low levels, in both cell lines. It is detectable using immunoprecipitation followed by Western blotting (see Bolanz et al., Mol Cancer Res, 2009; PMID: 19996302).

Second, previous studies have found elevated expression of TRPV6 in human breast cancer tissues (Zhuang et al., Lab Invest, 2001; PMID: 12480925. Dhennin-Duthille et al., Cell

Physiol Biochem, 2011; PMID: 22178934. Bolanz et al., Mol Cancer Res, 2009; PMID: 19996302) but with unclear underlying mechanism. Our present study was based on these clinical observations and aimed to examine how TRPV6 would worsen breast cancer. For this purpose, overexpressing WT or a GOF mutant TRPV6 in MCF-7 and MDA-MB-231 cells represents suitable approaches.

Would any Ca-permeable (WT or GOF mutant) channel have similar effects on breast cancer progression as TRPV6? We think that the answer would rather be 'no' because although Ca entry results in very complex changes in intracellular downstream signaling, TRPV6 has some specific, cancer-relevant, functions that other Ca-permeable channel may not have. For example, this study showed that TRPV6 interacts with PI3K and that this interaction is critical in mediating the effect of TRPV6 on cancer progression. Further, TRPV6 may have its intracellular interaction network that is distinct from those of other Ca-permeable channels and may also be important for regulating breast cancer progression.

Together, our study focused on a specific Ca channel (TRPV6) that when overexpressed in human breast cancer tissues is associated with a worse clinical outcome, and we used two cell lines to overexpress TRPV6 WT and mutant to model it, which is widely and similarly adopted by other cancer studies as *in vitro* models (see Fig. 4 in Raphaël et al., Proc Natl Acad Sci U S A, 2014; PMID: 25172921. Fig. 3 and 4 in Mu et al., Cancer Cell, 2003; PMID: 12676587. Fig. 6 in Takahashi et al., Cancer Cell, 2019; PMID: 29805077).

** See the Nature Portfolio author and referees' website at www.nature.com/authors for information about policies, services and author benefits

Communications Biology is committed to improving transparency in authorship. As part of our efforts in this direction, we are now requesting that all authors identified as 'corresponding author' create and link their Open Researcher and Contributor Identifier (ORCID) with their account on the Manuscript Tracking System prior to acceptance. ORCID helps the scientific community achieve unambiguous attribution of all scholarly contributions. You can create and link your ORCID from the home page of the Manuscript Tracking System by clicking on 'Modify my Springer Nature account' and following the instructions in the link below. Please also inform all co-authors that they can add their ORCIDs to their accounts and that they must do so prior to acceptance.

REVIEWERS' COMMENTS:

Reviewer #1 (Remarks to the Author):

The authors provided reasonable response to the comments, therefore I recommend acceptance.

I have one comment: "cancer cell progression" in the title and the abstract sound odd to me. I would change the title of the final published MS. For example:

Regulation of TRPV6 channel function by the S5/S6 helix interaction. Implications for cancer cell proliferation and migration.

Reviewer #2 (Remarks to the Author):

This is a revised version of a manuscript I reviewed in the spring of this year. The authors have adequately addressed the points raised in my previous review.

However, one point is still open: In the previous review I had recommended that the authors include the methods used to calculate the statistics in the figure legends. I did this because in the methods section of the previous manuscript the authors only mentioned that they used the Student's t-test. In the revised version of the manuscript, only the same information is given.

Now, t-tests are typically only used to compare two samples. In fact, in most of the data shown, more than two samples are compared. Accordingly, other tests must be applied that allow for multiple comparisons. I would have expected the authors to have recognised this and added it to the figure legends in the revised version. However, they have not done so. This point needs to be corrected for all comparisons in question.

Reviewer #3 (Remarks to the Author):

The authors have addressed all the issues raised.

Responses to reviewers' comments (in bold and start with ">>"; others are referees' original comments)

REVIEWERS' COMMENTS:

Reviewer #1 (Remarks to the Author):

The authors provided reasonable response to the comments, therefore I recommend acceptance.

I have one comment: "cancer cell progression" in the title and the abstract sound odd to me. I would change the title of the final published MS. For example:

Regulation of TRPV6 channel function by the S5/S6 helix interaction. Implications for cancer cell proliferation and migration.

>> Thanks for the suggestion. We have modified the title accordingly.

Reviewer #2 (Remarks to the Author):

This is a revised version of a manuscript I reviewed in the spring of this year. The authors have adequately addressed the points raised in my previous review.

However, one point is still open: In the previous review I had recommended that the authors include the methods used to calculate the statistics in the figure legends. I did this because in the methods section of the previous manuscript the authors only mentioned that they used the Student's t-test. In the revised version of the manuscript, only the same information is given.

Now, t-tests are typically only used to compare two samples. In fact, in most of the data shown, more than two samples are compared. Accordingly, other tests must be applied that allow for multiple comparisons. I would have expected the authors to have recognised this and added it to the figure legends in the revised version. However, they have not done so. This point needs to be corrected for all comparisons in question.

>> Thanks for the great suggestion. We performed one way ANOVA analysis for multiple groups comparisons, which has now been included in the Methods (page 21 line 579) to avoid repetition in the figure legends.

Reviewer #3 (Remarks to the Author):

The authors have addressed all the issues raised.

>> We would like to thank all reviewers' constructive comments during the revision.

** See the Nature Portfolio author and referees' website at www.nature.com/authors for information about policies, services and author benefits.